# What Is the Role of Body Composition Assessment in HCC Management?

**DOI:** 10.3390/cancers14215290

**Published:** 2022-10-27

**Authors:** Pompilia Radu, Maryam Ebadi, Aldo J. Montano-Loza, Jean Francois Dufour

**Affiliations:** 1Department of Visceral Surgery and Medicine, Inselspital, Bern University Hospital, University of Bern, 3008 Bern, Switzerland; 2Division of Gastroenterology & Liver Unit, University of Alberta, Edmonton, AB T6G 2X8, Canada; 3Department for BioMedical Research, Visceral Surgery and Medicine, University of Bern, 3008 Bern, Switzerland

**Keywords:** body mass composition, hepatocellular carcinoma, prognostic tool

## Abstract

**Simple Summary:**

Recent advances in evaluating nutritional status showed that body mass index is an inaccurate tool for assessing nutritional status. Furthermore, emerging data suggest that the body composition may be used as a non-invasive clinical tool with prognostic value in patients with hepatocellular carcinoma.

**Abstract:**

In the last decade, body composition (BC) assessment has emerged as an innovative tool that can offer valuable data concerning nutritional status in addition to the information provided by the classical parameters (i.e., body mass index, albumin). Furthermore, published data have revealed that different types of body composition are associated with different outcomes. For example, abnormalities of skeletal muscle, a common finding in cirrhotic and oncologic patients, are associated with poor outcome (i.e., high morbidity and high mortality). The disposition (visceral/subcutaneous adipose tissue) and radiodensity of adipose tissue proved to also be determinant factors for HCC outcome. Despite all the advantages, BC assessment is not part of the standard pre-therapeutic workup. The main reasons are the high heterogeneity of data, the paucity of prospective studies, the lack of a standard assessment method, and the interpopulation variation of BC. This paper aims to review the available evidence regarding the role of BC as a prognostic tool in the HCC population undergoing various therapies.

## 1. Introduction

The prognosis of patients with hepatocellular carcinoma (HCC) is very poor, despite the cumulative advances in diagnostic procedures and treatment strategies, placing this tumor as the second-leading cause of cancer-related deaths worldwide [1]. Currently, we are confronted with a huge knowledge gap about prognostic factors, and the daily clinic experience confirms that besides hepatic functional reserve and tumor stage, there are other factors that need to be considered. 

A growing number of studies have concluded that the evaluation of body composition (BC) in addition to an accurate assessment of nutritional status, might also serve as a potential prognostic factor in cirrhosis and oncologic patients [2,3,4,5].

Early identification of these changes might become an important step in HCC patient management. However, we are not adequately using this indicator due to several drawbacks, such as data heterogeneity, the wide variety of diagnostic tools and thresholds, population heterogeneity, and the lack of prospective studies.

Notably, EASL guidelines recommend using specific nutritional support strategies in managing patients with HCC [6]. Furthermore, different physical exercises have been suggested as an adjuvant measure in order to reduce muscle atrophy and stimulate weight loss in patients with HCC.

In the following paragraphs, we aimed to review studies evaluating the role of different body composition phenotypes in HCC patients and the outcome after different treatments, such as loco-regional, surgery, transplant, and chemo/immunotherapy. 

## 2. Current Methods Used to Evaluate Body Mass Composition

Currently, there is high variability in the method used to evaluate the BC. Figure 1 illustrates the equipment, advantages, and disadvantages of the most commonly-used body composition assessment methods.

**Magnetic resonance imaging** (**MRI**) **and computed tomography** (**CT**) are considered the gold standard techniques for the assessment of BC. Based on radiodensity measured in Hounsfield units (HU), each component of BC can be identified (i.e., skeletal muscle −29 HU to 150 HU; visceral −150 to −50 HU; and subcutaneous adipose tissue −190 to −30 HU) [7,8]. A single abdominal CT image at the third lumbar vertebra (L3) is an accurate indicator of the total body musculature and adipose tissue [9].

**Bioelectrical impedance analysis** (**BIA**) is one of the most used methods in clinical practice and research. The BC is evaluated based on the rate at which electrical current travels through the body. The main limitations of BIA are described in overweight and obese patients as the result of (1) an inadequate BIA equation that are developed in normal-weight subjects, (2) a different body water distribution in severe obesity [10].

**Dual-energy X-ray** is one of the most popular methods used to evaluate body composition. The physical principle beyond DXA is that of X-ray transmission and attenuation through a body at two different energy levels [11]. After the whole body is scanned, based on the attenuation of the X-ray beam, the body is split into two compartments: bone and soft tissue. After specific soft tissue algorithms are applied, the DXA scanner is able to partition the human body into the fat, lean, and bone components [12]. Of note, the soft tissue measurements can only be made in regions of the body where no bone is present. In clinical practice, appendicular skeletal muscle mass (ASM), which is the sum of the lean muscle mass of both arms and legs, is used to evaluate the musculature compartment.

## 3. The Impact of Skeletal Muscle Abnormalities on HCC Outcome

Several abnormalities of the musculature compartment, such as low skeletal muscle mass (LSMM); sarcopenia (reduction in muscle mass and function); sarcopenic obesity (simultaneous loss of skeletal muscle and gain of adipose tissue); myosteatosis (increased proportion of intermuscular and intramuscular fat); and cachexia (an irreversible process that leads to the loss of both muscle mass and fat tissue) have been reported in cirrhotic and oncologic patients, including HCC. All these changes are the result of complex processes, malnutrition, decreased physical activity, increased inflammatory response, alterations in the hormonal milieu (i.e., low testosterone levels), and an imbalance between anabolic and catabolic metabolisms [13,14]. 

Despite the clear negative implication of musculature abnormalities (sarcopenia, myosteatosis, and sarcopenic obesity) on the outcome in cirrhosis and HCC setting, there is limited data on the pathophysiologic mechanism [1,15,16,17,18]. Recent studies have revealed that besides its well-known functions, skeletal muscle is an endocrine organ that can modulate different organs, including the immune system [19,20]. In response to muscular activity, myocytes produce myokines. Via several myokines (i.e., IL-6, IL-7, and IL-15), the skeletal muscle modulates the immune system [20,21]). On the other hand, the immune system via proinflammatory cytokine (i.e., tumor necrosis factor alpha (TNF), IL-6, and transforming growth factor ß (TGF ß)) activates the expression of several genes, such as the E3 ubiquitin ligase muscle RING finger containing protein 1 (MURF1) and the muscle atrophy F box protein (MAFbx), which are responsible for the catabolic state [22] (Rao) 2022.

Immune cells in turn critically influence muscle mass and function. 

### 3.1. The Effect of Sarcopenia on the Outcome of HCC

#### 3.1.1. Liver Transplantation

According to the updated Barcelona Clinic Liver Cancer (BCLC) recommendation, liver transplantation (LT) is the treatment of choice for patients with early-stage HCC who are unsuitable for resection, as well as for BCLC B patients who meet extended liver transplant criteria [23]. The risk of HCC recurrence (8–20%), the organ shortage, the number of deaths on the waiting list, and the risk of perioperative complications are the main concerns in these patients [24]. 

Sarcopenia proved to be a negative factor after LT in cirrhotic patients, but due to conflicting results, the impact of muscle sarcopenia on the post-transplant HCC outcome remains uncertain. Hamaguchi et al. reported a significant association between low psoas muscle mass index (PMI) (HR = 3.635, 95% CI = 1.896–7.174, *p* < 0.001) and post-transplant survival [25]. In 2013, Meza-Junco et al. found in a retrospective study that the mortality risk was two times higher in sarcopenic than nonsarcopenic patients [26]. Interestingly, in the same updated cohort, the results were not confirmed in terms of mortality; however, the authors found that sarcopenia was predictive for a longer recovery period after transplantation (i.e., longer hospital stays, a higher risk of perioperative bacterial infections after LT) [27]. Valero V et al. failed to find a significant association between sarcopenia and overall survival (HR = 1.23; *p* = 0.51) [28]. However, the authors also reported a higher rate of infections (i.e., sepsis, pneumonia), postoperative complications, and longer recovery after LT in the sarcopenic group [28,29]. In an Asian cohort (HCC beyond the Milan criteria), Kim YR found that sarcopenia was independently associated with tumor recurrence after LT, which was the main cause of death in this group [30]. In line with these findings, Beumer concluded—based on a multicenter cohort (18 centers, *n* = 889 assessed to LT for HCC beyond the Milan criteria)—that a higher muscle mass contributes to better long-term survival [31]. 

Another controversial point concerns the reversal of sarcopenia after LT. While Montano-Loza found that more than 28% of transplanted patients have improved their muscle health, Bhanji et al. found that only a minority of transplanted patients demonstrated an improvement [27,32]. Moreover, in around 25% of the transplanted patients, sarcopenia continued to accentuate in the first year following the transplant [32]. The etiology of the progressive sarcopenia seems to be multifactorial and includes a persistent catabolic state, immunosuppression (i.e., mTOR, calcineurin inhibitors), corticosteroid use, a reduced physical activity due to long recovery, renal failure, and possibly a recurrent liver disease [33]. 

An improved nutrition has been reported to significantly improve the overall survival in pre-transplant sarcopenic patients [34,35]. Interestingly, in one of these studies, this effect was not found in patients with normal/high skeletal muscle mass (*p* = 0.550) [34]. Moreover, similar to previous studies, neither the preoperative Child–Pugh classification nor the Model for End-Stage Liver Disease (MELD) score were significantly different between patients with or without nutritional therapy in sarcopenic patients [34]. This confirms that despite being indisputable tools in evaluating liver function, the Child-Pugh and MELD scores cannot accurately assess the nutritional and functional status. Adding sarcopenia to MELD (MELD-sarcopenia and MELD-psoas) has been associated with improved accuracy in the prediction of mortality for patients with cirrhosis, particularly in patients with MELD ≤ 15 [36,37].

#### 3.1.2. Liver Resection

The role of liver resection in HCC management has increased over the last decade. To decrease the risk of complications, recurrence, and mortality, patients are assigned to surgery based on good liver function (i.e., ALBI score, Child-Pugh score, and MELD score), lack of portal hypertension, and tumor burden (i.e., BCLC criteria) [23]. However, recurrence and mortality rates remain relatively high. Thus, a more individualized approach is mandatory. In an Asian HCC cohort (40.3% sarcopenic patients) assigned to hepatic resection, Harimoto et al. reported a significantly worse 5 years OS for sarcopenic patients compared with the non-sarcopenic ones (71% vs. 83·7%) [38]. Later, other authors confirmed that sarcopenia was a strong and independent prognostic factor for mortality after curative HCC resection [39,40]. In a prospective study, Dello SA et al. reported that sarcopenic patients (67.5%, *n* = 20) undergoing liver resection had a smaller normal liver volume (TFLV) than in the non-sarcopenia group [1396 mL (TFLV: 1129–2625 mL) and 1840 mL (TFLV: 867–2404 mL), respectively; *p* < 0.050] [41]. This might be one of the explanations for the increased risk of postoperative complications in sarcopenic patients [42]. In addition, Takagi et al. found a positive correlation between the presence of microvascular invasions (*p* = 0.003) and the tumor stage (*p* = 0.015) [39]. As a confirmation of these findings, Yabusaki et al. reported that sarcopenia was a significant independent risk factor for HCC recurrence (HR = 1.6; 95% CI 1.1–2.5; *p* = 0.02) in patients with normal BMI [43]. In a meta-analysis by Xu L, preoperative sarcopenia was significantly associated with mortality and additionally, in the Asian cohorts but not in Caucasian ones, sarcopenia was significantly associated with tumor size [18]. The differences in the cut-off values for sarcopenia and the lower number of Caucasian studies were considered responsible for these discrepancies. 

#### 3.1.3. Local Ablative Therapy (RFA and MWA)

Concerning the impact of sarcopenia on the outcome (i.e., recurrence and survival) after ablative methods (radiofrequency ablation (RFA)/microwave ablation (MWA)), there is a paucity of data. 

In a prospective study, Fujiwara et al. showed that in the group of HCC patients (*n* = 515) staged as BCLC 0/A and assigned to RFA, sarcopenia was associated with a higher risk for HCC recurrence, while all three factors were significantly associated with higher mortality [44]. In an Asian cohort, Yuri et al. found sarcopenic patients assigned to RFA, irrespective of the Child–Pugh stage, survived significantly less than those without sarcopenia (51.5% vs. 86.5%). Sarcopenia remained an independent predictor in the multivariate analysis (HR of 6.867, 95% CI 3.498–14.425) [45]. Of note in this study, there was no difference between the recurrence rate of sarcopenic and non-sarcopenic patients (*p* = 0.5077). Recently, Yeh et al. reported that in their cohort assigned to RFA, pre-sarcopenia was the prognostic of a worse 5 year OS rate compared with non-sarcopenic ones (44.1% vs. 68.9%), but not of recurrence [46]. Contrary, Kamachi S., found sarcopenia as an independent risk factor for HCC recurrence in patients assigned to curative therapy (resection 35 (57%) and RFA 26 (43%)) (*p* = 0.03) [47]. One of the explanations for the association/lack of association between sarcopenia and HCC recurrence might be the tumor size, which is known to be a risk factor for recurrence. Table 1 summarizes the last studies that have evaluated the relation between changes in the musculature compartment and the outcome of HCC patients treated curatively.

#### 3.1.4. Chemoembolization

The response to trans-arterial chemoembolization (TACE) is heterogeneous and the appropriate selection of patients who will benefit from TACE in terms of overall survival without hepatic decompensation is an unresolved clinical issue. In this setting of patients, the reported data about the impact of sarcopenia is conflicting. 

In two Asian cohorts, the authors found no significant link between the muscle mass at baseline and the clinical outcome [54,55]. On the contrary, in two European cohorts, Dodson et al. and Loosen et al. showed that pre-interventional sarcopenia was an independent predictor for an unfavorable outcome (respectively HR 1.84, 95% CI 1.03–3.64, *p* =  0.04 and HR 2.876, 95% CI 1.044–7.922, *p =* 0.04) [56,57]. In the study conducted by Loosen, sarcopenia had no impact on the response to TACE therapy [57].

#### 3.1.5. Systemic Therapy

Table 2 presents the studies published in the last 5 years that have investigated the impact of sarcopenia/sarcopenic obesity on systemic therapy. 

##### Tyrosine Kinase Inhibitors

Sarcopenia has been reported as a prognostic tool for poor outcomes in patients treated with sorafenib. In the Asian population treated with sorafenib, sarcopenia was associated with the occurrence of early dose-limiting toxicities and reduced overall survival rate [58,61,63,65]. The results were mirrored by a European multicentric retrospective study that confirmed the reduced OS (63 vs. 32 weeks, HR 1.69, *p* = 0.02), the reduced duration of treatment (25.8 vs. 12.3 weeks, HR 1.75, *p* =  0.044), and the higher sorafenib-related toxicity (adverse events grade 3 and 4, 62% vs. 40%, *p* = 0.04) [62]. The antiangiogenic effect of sorafenib can directly inhibit protein synthesis, which leads to a progressive loss of musculature during the therapy with sorafenib [59]. The negative variation of transverse psoas muscle thickness (measured on CT at the level of the umbilicus)/height (TPMT/m^2^) before and after sorafenib treatment, was an independent factor for a low OS (HR = 2.27, *p* = 0.02) [60]. The cut-off value for the difference in TPMT/m^2^ was 0.59 mm/m^2^. 

##### Immunotherapy

Recently, emerging evidence has indicated sarcopenia as an independent, unfavorable prognostic factor in oncologic patients receiving immunotherapy [17,66,67]. In addition to their role in T cell exhaustion, TGF-ß and Il 6 seem to be responsible for the impaired response to immunotherapy [67,68]. In an Asian HCC population assigned to immunotherapy (*n* = 138), sarcopenia was present in 46.5% of patients and it was associated with poor survival outcomes (HR = 2.09, *p* = 0.003) [16]. On the contrary, in an American HCC cohort (*n* = 57) assessed to immunotherapy, sex-specific sarcopenia showed a trend of worse OS (HR: 1.71, 95%) but was not statistically significant (*p* = 0.215) [64]. The discrepancy might be explained by the relatively low number of patients included in the American cohort. 

#### 3.1.6. Myosteatosis

Currently, there is no standard cutoff to define myosteatosis, and as a direct result, this impacts the prevalence and associations with clinical outcomes. 

However, published data indicate that myosteatosis may be considered a negative factor for survival in pre- and post-LT, irrespective of the presence of HCC [51,69]. Moreover, a large retrospective study conducted in Japan reported that myosteatosis is closely related to the recurrence of HCC in patients receiving curative HCC treatment [44]. Recently, the data of a European cohort supported the importance of myosteatosis over sarcopenia. Thus, patients with myosteatosis had a higher incidence of major postoperative complications (*p* = 0.007) and tendentially worse survival (median survival: 41 vs. 60 months, *p* = 0.223) [52]. In two Asian cohorts assigned to immunotherapy (discovery cohort *n* = 111, validation cohort *n* = 27), myosteatosis was an independent predictor of progression-free survival (*p* = 0.014) and OS (*p* = 0.007), while sarcopenia was independent for OS (*p* = 0.007) in these patients [16].

#### 3.1.7. Sarcopenic Obesity

Sarcopenic obesity was defined as the presence of sarcopenia in obese patients. Its presence was linked with a poor prognosis in HCC patients (Table 1). The cutoffs for this profile are varying within the population (i.e., Caucasian, Asian). In some Asian studies, obesity was diagnosed based on VAT area at the level of the umbilicus, and not on BMI cutoffs due to its inability to discriminate between fat and fluid retention. 

Patients with a sarcopenic obesity profile have a proinflammatory milieu (high levels of leptin, chemerin, and resistin, tumor necrosis factor (TNF)-α, interleukin (IL)-6, interferon (INF)-γ), and a decreased concentration of adiponectin and IL-15, which contributes to the progression of HCC [12]. In a cohort of 465 HCC patients assigned to resection, Kobayashi et al. reported that pre-operative sarcopenic obesity was an independent predictor for mortality and HCC recurrence [50]. On the contrary, Kroh et al. reported that sarcopenic obesity found no detrimental effect of sarcopenia/sarcopenic obesity on postoperative survival in HCC patients assessed for surgery [49]. 

## 4. Adipose Compartment

Recent findings indicate that adipose tissue is actively involved in the modulation of endocrine function, immunity (i.e., leptin and adiponectin), angiogenesis (i.e., angiopoietin-2 (Angpt2), the vascular endothelial growth factor (VEGF), leptin, and adiponectin) [70,71]. Based on the distribution, adipose tissue was divided into subcutaneous adipose tissue (SAT) and visceral adipose tissue (VAT), components with different functions that may explain the favorable effects of SAT and the negative effects of VAT frequently reported in cancer patients. Furthermore, recent studies have shown that not only does the volume of each component have an impact on outcomes in cirrhotic and oncologic patients (HCC, sarcoma, and myeloma) but also the quality of fat, evaluated by radiodensity (Hounsfield units). Thus, the high radiodensity of the adipose tissue (the switch toward fewer negative values) has been associated with higher risks of death among cirrhotic patients and in patients with advanced cancer, including HCC, metastatic CRC, and extremity sarcoma [69,72,73,74]. The switch toward a high radiodensity is the result of several events, such as a reduction in the lipid content, the browning of the white adipose tissue, the increase in the fibrous tissue, and more adipose tissue inflammation [75,76,77].

### 4.1. Subcutaneous Adipose Tissue

In patients with a chronic catabolic status, especially in oncologic ones, adipose tissue lipolysis augments, whereas adipogenesis decreases [78,79]. SAT is the main provider of energy in hypercatabolic states (i.e., cirrhosis and oncological events), playing a protective role against increased energy exhaustion. In addition, it stimulates the insulin response, glucose and lipid metabolism, and the immune response via adipokines and primarily leptin [69]. These observations are reflected in the results of several studies that have linked the low volume of SAT with a poor prognosis, irrespective of the HCC therapy (Table 3). 

In the study conducted by Kobayaski, a high SAT volume was associated with improved survival outcomes of HCC patients treated with transcatheter intra-arterial therapies [80]. A rapid depletion of SAT during sorafenib treatment (∆SATI: <−32.51 for women and <−15.49 for men) seemed to be an early sign of malnutrition and was associated with poor survival in HCC patients [81].

Despite the variation of the cutoffs, a shift toward less negative values of SAT was found to be a negative prognostic factor for the outcome in cirrhotic and oncological patients irrespective of the tumor stage [69,72].

### 4.2. Visceral Adipose Tissue (Volume and Radiodensity)

Contrary to the favorable effects of an increased SAT, an increased VAT has been reported as a negative factor for HCC patients.

In a NASH Asian cohort, Ohki T. et al. found that high VAT (>130 cm^2^ in males, >90 cm^2^ in females) was an independent risk factor of HCC recurrence after radiofrequency ablation [82]. Fujiwara N. et al. found that increased VAT, sarcopenia, and intramuscular fat deposition could predict mortality in patients with HCC [44]. Cirrhotic male patients with a VAT index ≥65 cm^2^/m^2^ had a higher risk of HCC pre-LT and a higher risk of HCC recurrence post LT (HR 2.21; 95% CI, 1.45–3.38; *p* < 0.001 and HR 5.17; 95% CI 1.15–23.21; *p* = 0.03, respectively) [83]. In an Asian cohort assigned to hepatectomy for HCC, Hamaguchi Y. found that a preoperative high visceral to subcutaneous adipose ratio (>1.325 for males and >0.710 for females) was a significant risk factor for mortality (HR = 1.566, *p* < 0.001) and HCC recurrence (HR = 1.329, *p* = 0.020) [51]. From a pathophysiological point of view, the excess accumulation of VAT was linked to an increased expression of the proinflammatory cytokines (i.e., tumor necrosis factor-a, IL-6, leptin, and monocyte chemoattractant protein-1) and a decrease in the anti-inflammatory ones (i.e., adiponectin) [80,84]. All these changes in elderly or sarcopenic populations have been shown to lead to immunosenescence, particularly of the natural killer lymphocytes involved in innate immunity [85]. Moreover, VAT is associated with insulin resistance, characterized by elevation of both insulin-like growth factor-1 (IGF-1) and insulin levels, associated with an increased autophagy, which promoted HepG2 cell survival and metastasis, leading to a worse prognosis for HCC patients [86].

Similar to SAT, a high radiodensity of VAT (i.e., morphological features of atrophy and chronic inflammation) was associated with a negative outcome in HCC patients. High VAT (>−85 HU) radiodensity proved to be associated with both an increased mortality (HR 2.01, 95% CI 1.14–3.54, *p* = 0.02) and severe adverse events in patients treated with selective internal radiation therapy [87]. The positive association between a low VAT radiodensity (<−89.1 HU) and tumor response (complete response and partial response) (HR: 1.035, 95% CI: 1.014–1.058, *p* = 0.001) to TACE was recently reported by Li Q [88]. Table 4 presents the studies that have evaluated the relation between VAT and HCC outcome published in the last 5 years.

## 5. Conclusions

In conclusion, nutritional status seems to play an important role in the outcome of HCC patients, irrespective of the assigned therapy. BC could offer additional information concerning the nutritional status of those provided by the classical markers or scores (i.e., albumin, BMI, Child–Pugh, and MELD). In order to use all the data provided by the BC analysis and to develop reliable prognostic tools, several issues need to be resolved over the next few years. Current issues can be addressed by gaining a better understanding of the pathophysiological mechanisms, conducting additional prospective studies, as well as establishing standardized methods, protocols, and cutoffs for BC assessment.

## Figures and Tables

**Figure 1 cancers-14-05290-f001:**
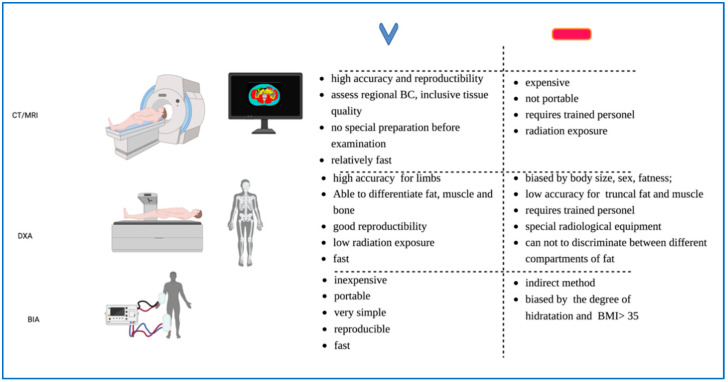
The most common methods for assessing body composition (equipment, advantages, and disadvantages).

**Table 1 cancers-14-05290-t001:** Studies evaluating the impact of skeletal musculature changings on HCC patient outcomes assigned to curative therapies published in the last 5 years.

First Author and Year	Study Design	Cohort Characteristic	Method Used Cutoffs Used	Conclusion
Liver transplantation				
Kim Y.R. et al., 2018 [30]	Retrospective	Asian 92 HCC beyond MC	CT-based segmentation at L3 PMI cutoff <15.5 mm/m	Sarcopenia risk factor for recurrence (HR = 9.49 95% CI 1.18–76.32 (*p* = 0.034))
Chae M.S. et al., 2018 [48]	Retrospective	Asian 408 (46.8% HCC)	Preoperative CT-based segmentation at L3 PMI at L3	A PMI decrease ≤−11.7% between the before surgery and 7th day post LT was an independent predictor of patient mortality after LT
Acosta L. et al., 2019 [29]	Retrospective	North America 163	Preoperative CT-based segmentation at L3 SMI 52.4 cm^2^/m^2^ in males and SMI 38.5 cm^2^/m^2^ in women	Patients in the lowest quartile of the SMI were associated with 70% increased risk of prolonged length of stay in this cohort
Beumer R. et al., 2022 [31]	Retrospective multicenter	European 889 HCC beyond MC	Preoperative CT-based segmentation at L3 In women, SMI: 37 cm^2^/m^2^ for BMI < 25 kg/m^2^, 42 cm^2^/m^2^ for BMI ≥ 25 kg/m^2^ In men SMI 45 cm^2^/m^2^ for BMI < 25 kg/m^2^, 51 cm^2^/m^2^ for BMI ≥ 25 kg/m^2^	Patients with higher muscle mass had a better long-term survival
Liver resection				
Kroh A. et al., 2018 [49]	Retrospective	Asian 70 patients	Preoperative CT-based segmentation at L3 Sarcopenia In men, SMI < 43 for BMI < 25 kg/m^2^, SMI < 53 for BMI > 25 kg/m^2^ In women, SMI < 41 irrespective of the BMI Obesity was defined based on the top two body fat percentage quintiles for men and women, respectively	Sarcopenia, obesity, and sarcopenic obesity were not risk factors for poor postoperative survival in this study
Kobayashi A. et al., 2019 [50]	Retrospective	Asian 465 patients	Preoperative CT-based segmentation at L3 SMI cutoff <40.31 cm^2^/m^2^ for men <30.88 cm^2^/m^2^ for women Obesity: visceral adipose tissue area was >100 cm^2^ in both males and women	Sarcopenic obesity risk factor for mortality (HR = 2.504, *p* = 0.005) and recurrence of HCC (HR = 2.031, *p* = 0.006)
Hamaguchi Y. et al., 2019 [51]	Retrospective	Asian 606 patients	Preoperative CT-based segmentation at L3 Low SMI cutoff: <40.31 cm^2^/m^2^ for men and <30.88 cm^2^/m^2^ for women High VSR cutoff: >1.325 for males and >0.710 for women High IMAT cutoff: >–0.358 for males and >–0.229 for women	A high VRS, low SMI, and high IMAC contributed to an increased risk of death (*p* < 0.001) and HCC recurrence (*p* < 0.001) in an additive manner
Meister F. et al., 2022 [52]	Retrospective	European 100 patients	Preoperative CT-based segmentation at L3 SM-RA < 41 HU for patients with BMI up to 24.9 kg/m^2^ and <33 HU for patients with a BMI ≥ 25 kg/m^2^	Myosteatosis was as an independent risk factor for perioperative morbidity (HR: 6.184, 95% CI 1.184–32.305, *p* = 0.031) Myosteatotic vs. non-myosteatotic (41 months vs. 60 months, *p* = 0.223)
Jang H.Y. et al., 2021 [53]	Retrospective	Asian 160 patients	Preoperative CT-based segmentation at L3: PMI cut-off: <3.33 for male, 2.38 for female VATI cut-off ≥30.39 for male, >44.70 for women	Sarcopenia and high VATI was associated with poor OS but not recurrence-free survival PMA did not predict OS
RFA/MWA				
Yuri Y. et al., 2017 [45]	Retrospective	Asian 182	CT-based segmentation at L3 PMI cut-off: 6.36 cm^2^/m^2^ for men and 3.92 cm^2^/m^2^ for women	Sarcopenia was associated with a reduced OS with no effect on recurrence

Abbreviation: LT, liver transplant; RFA, radiofrequency ablation, MWA, microwave ablation; HCC, hepatocellular carcinoma; MC, Milan criteria; BMI, body mass index, PMI, psoas muscle index; L3-SMI, third lumbar vertebrae–skeletal muscle index; IMAT, intermuscular adipose tissue; VATI, visceral adipose tissue index, SMI, skeletal muscle index; PMA, psoas muscle attenuation; PMI, psoas muscle index; SM-RA, skeletal muscle radiation attenuation; TPA, total psoas area; and VSR, visceral-to-subcutaneous adipose tissue area ratio; HR, hazard ratio; CI, confidence interval; OS, overall survival.

**Table 2 cancers-14-05290-t002:** Summary of published studies, in the last five years, evaluating the impact of skeletal musculature changes on patient outcomes undergoing systemic therapy (tyrosine kinase inhibitors and immunotherapy).

First Author and Year	Study Design	Cohort Characteristic	Method Used Cutoffs Used	Major Finding
Nishikawa H. et al., 2017 [58]	Retrospective	Asian 232	CT-based segmentation at L3 Cut off L3-SMI ≤ 36.2 cm^2^/m^2^ for male; ≤29.6 cm^2^/m^2^ for women	Sarcopenia is an independent predictor of low OS (HR 0.365; *p* < 0.0001) Sarcopenic patients had a lower rate of objective response rate and disease control rate
Hiraoka A. et al., 2017 [59]	Retrospective	Asian 93	CT-based segmentation at L3 Cut off PSI: <4.24 cm^2^/m^2^ for male; <2.50 cm^2^/m^2^ for female	Sarcopenia is an important negative factor in patients treated with sorafenib
Yamashima M. et al., 2017 [60]	Retrospective	Asian 40	CT-based segmentation at L3 TPMT was evaluated prior to treatment initiation and after 1–3 months of treatment ΔTPMT/height < 0.59 mm/m^2^	Patients with mild muscle atrophy exhibited a significantly longer OS compared with patients with severe muscle atrophy (*p* = 0.045)
Takada H. et al., 2018 [61]	Retrospective	Asian 214	CT-based segmentation at L3 Cut off L3-SMI <42 cm^2^/m^2^ for male: <38 cm^2^/m^2^ for women	Pretherapy sarcopenia in patients with two or less negative prognostic factors is an important negative prognostic factor (HR 1.6; *p* = 0.047)
Antonelli G. et al., 2018 [62]	Retrospective	Europe 96	CT-based segmentation at L3 within 30 days from treatment start Cut off L3-SMI < 53 cm^2^/m^2^ if BMI > 25 and < 43 cm^2^/m^2^ if BMI < 25 for men <41 cm^2^/m^2^ for women	Sarcopenia is associated with reduced survival and reduced duration of sorafenib
Imai K. et al., 2019 [63]	Retrospective	Asian 61	CT-based segmentation at L3 Cut off L3-SMI < 42 cm^2^/m^2^ for male: <38 cm^2^/m^2^ for women ΔL3-SMI > −5.73 cm^2^/m^2^/120 days ΔSFMI > −5.33 cm^2^/m^2^/120 days ∆VFMI > −3.95 cm^2^/m^2^/120 days	Rapid depletions in subcutaneous fat mass and skeletal muscle mass after the introduction of sorafenib indicate a poor prognosis
Immunotherapy				
Akce M. et al., 2021 [64]	Retrospective	American 57	Pretreatment CT at L3 level SMI cut-off: 43 cm^2^/m^2^ for males and 39 cm^2^/m^2^ for women	Sex-specific sarcopenia does not predict OS, whereas baseline BMI and NLR together may predict OS in advanced HCC patients treated with anti-PD-1 antibody
Chen B.B. et al., 2022 [16]	Retrospective	Asian 138	Pretreatment CT at L3 level SMD cutoff: <41 HU for BMI < 25 kg/m^2^, and <33 HU in for BMI ≥ 25 kg/m^2^ SMI cut-off: 40.8 cm^2^/m^2^ for men and 34.9 cm^2^/m^2^ for women Sarcopenic obesity: sarcopenia in patients with BMI > 25 kg/m^2^	Sarcopenia and myosteatosis had a negative impact in patients who received immunotherapy for advanced HCC

Abbreviation: CT, computer tomography; L3-SMI, third lumbar vertebrae–skeletal muscle index; HCC, hepatocellular carcinoma; HU, Hounsfield units; SMI, skeletal muscle index; SMD, skeletal muscle density; ∆L3SMI, change in third lumbar vertebra skeletal muscle index; ∆SFMI, change in subcutaneous fat mass index; ∆VFMI, change in visceral fat mass index; ΔTPMT/height, changes in transverse psoas muscle thickness per height; BMI, body mass index; PSI, psoas muscle area; NLR, neutrophil-to-lymphocyte ratio; HR, Hazard ratio; OS, overall survival.

**Table 3 cancers-14-05290-t003:** Studies assessing the correlation between subcutaneous adipose tissue and the outcome of HCC patients published in the last 5 years.

First Author and Year	Study Design	Cohort Characteristic	Method Used Cutoffs Used	Conclusion
Kobayashi T. et al., 2018 [80]	Retrospective	Asian 100 HCC patients assigned to TACE	Preoperative CT-based segmentation at L3	High SAT volume is associated with better survival outcomes in HCC patients treated with TACE
Imai K. et al., 2019 [81]	Retrospective	Asian 61	CT-based segmentation at L3 Cutoff L3-SMI < 42 cm^2^/m^2^ for male: <38 cm^2^/m^2^ for women ΔL3-SMI > −5.73 cm^2^/m^2^/120 days ΔSFMI > −5.33 cm^2^/m^2^/120 days ∆VFMI > −3.95 cm^2^/m^2^/120 days	Rapid depletions of subcutaneous fat mass and skeletal muscle mass after the introduction of sorafenib indicate a poor prognosis

Abbreviations: CT, computer tomography; L3-SMI, third lumbar vertebrae–skeletal muscle index; HU, Hounsfield units; SAT, subcutaneous adipose tissue; and SMI, skeletal muscle index; ∆L3-SMI, change in third lumbar vertebra skeletal muscle index; ∆SFMI, change in subcutaneous fat mass index; ∆VFMI, change in visceral fat mass index; TACE, transarterial chemoembolization

**Table 4 cancers-14-05290-t004:** Studies assessing the correlation between visceral adipose tissue and the outcome of HCC patients published in the last 5 years.

First Author and Year	Study Design	Cohort Characteristic	Method Used Cutoffs Used	Conclusion
Parikh et al., 2018 [89]	Retrospective	Asian 124 patients pre-LT	Multifrequency BIA	IMAT (HR = 3.898, 95% CI = 2.025–7.757, *p* < 0.001] and low PMI (HR = 3.635, 95% CI = 1.896–7.174, *p* < 0.001) were independent risk factors for death after LDLT
Montano-Loza et al., 2018 [83]	Retrospective	Canadian 678 patients (289 with HCC) pre-LT	CT-based segmentation at L3	VATI ≥ 65 cm^2^/m^2^ independent risk factor for HCC in male patients with cirrhosis and for recurrence of HCC after LT
Hamaguchi et al., 2019 [51]	Retrospective	Asian 606 patients	Preoperative CT-based segmentation at L3 SMI cutoff: <40.31 cm^2^/m^2^ for men and <30.88 cm^2^/m^2^ for women	A high VRS (HR = 1.329, *p* = 0.020), low SMI, and high IMAT contributed to an increased risk of death (*p* < 0.001) and HCC recurrence (*p* < 0.001) in an additive manner
Ebadi et al., 2020 [87]	Retrospective	Canadian 89 HCC patients assess to SIRT	CT-based segmentation at L3 VAT cutoff: –85 HU	VAT ≥ –85 HU had a 2× higher risk of mortality (HR 2.01, 95% CI 1.14–3.54, *p* = 0.02) compared with their counterpart
Li Q et al., 2020 [88]	Retrospective	Asian 192 intermediate stage HCC patients assigned to TACE	CT-based segmentation at L3 VAT cutoff: −89.1 HU	VAT < −89.1 HU associated with better OS and PFS (25.1 mo, 95% CI: 18.1–32.1 vs. 17.6 mo, 95% CI: 16.3–18.8, *p* < 0.0001, 15.4 mo, 95% CI: 10.6–20.2 vs. 6.6 mo, 95% CI: 4.9–8.3, *p* < 0.0001, respectively)

Abbreviations: HCC, hepatocellular carcinoma; Pre LT, pre liver transplantation; LT, liver transplantation; TACE, Transarterial chemoembolization; SIRT, Selective Interne radio-therapie CT, computer tomography; BIA, bioelectrical impedance analysis; L3-SMI, third lumbar vertebrae–skeletal muscle index; HU, Hounsfield units; IMAT, Intermuscular adipose tissue, VAT, visceral adipose tissue; VATI, visceral adipose tissue index; PMI, psoas muscle index; SMI, Skeletal muscle mass index; HR, Hazard ratio; CI, confidence interval; mo, months.

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
