# Peer review of "What Is the Role of Body Composition Assessment in HCC Management?"

_cancers, 2022, doi:10.3390/cancers14215290_

Round 1
Reviewer 1 Report
This review summarizes the role of sarcopenia (the most part) and fatty tissue (minor part) in HCC patients.
Comments
1. A diagram of assessment of body composition (as the title suggested) in HCC patients regarding future risk is recommended. The authors can combine fat and muscle into one diagram.
2. Can visualized presentations (a forest plot) be added on the side of Table 1-4?
3. More specific statements derived from your review should be included in the abstract.
Reviewer 2 Report
Pompilia Radu et al. wrote a comprehensive review on how body composition assessment could affect HCC management and how could physicians could possibly use BC as a potential prognostic factor to better manage patients with HCC.
The authors reviewed a number of studies and systematically summarized the association of BC with the outcomes of HCC patients after different treatments, such as loco-regional, surgery, transplant, and chemo/immunotherapy. Although BC information such as sarcopenia and VAT could provide additional information to help predict the poor prognosis and overall survival of HCC patients, there are no standardized protocols to take BC data into account in terms of combining it to develop a new prognostic tool. All in all, this manuscript is well-written and properly structured. It provides an evidence-based review to suggest how BC data could be interpreted in terms of evaluating the outcomes of patients of HCC
Minor point.
1. In the 3.1.5 systemic therapy section, the authors seem to skip this section discussing the association of BC (such as sarcopenia and VAT) with the outcome of HCC patients.
Author Response
We would like to thank the reviewer for careful and thorough reading of this manuscript and for the thoughtful comments and constructive suggestions, which help to improve the quality of this manuscript. Our response follows:
Reviewer 2
In the 3.1.5 systemic therapy section, the authors seem to skip this section discussing the association of BC (such as sarcopenia and VAT) with the outcome of HCC patients.
Answer: Thank you very much for pointing out this aspect. In the manuscript, we discussed the association of sarcopenia and systemic therapy, but it was an issue with numbering the subsection. We updated the number of each subsection. Now data concerning the association of sarcopenia with immunotherapy can be found in subsection 3.1.5.1 (Tyrosine kinase inhibitors) and 3.1.5.2 (immunotherapy) respectively.
Reviewer 3 Report
This is a well-written manuscript covering a crucial topic in the modern oncology regarding nutritional status and its effect on various therapies.
Minor English spelling mistakes were mentioned throughout the text and should be corrected.
In addition, advanced and metastatic HCC is usually accompanied by cachexia. Cachexia itself is a depletion tool for various compartments and impacting to decreased body composition. And the more advanced tumour the more profound is weight loss (at least in case of HCC). Could you please add a few sentences if there are any predisposing genetic differences for cachexia development in some patients.
Also, the advanced stages of liver cancer can be also associated with jaundice, ascites, transaminitis. Please add a few statements regarding their role in body composition balance shifting (e.g. bilirubin as a product of heme destruction is toxic if increased in circulation leads to possible renal damage leading to lower filtration thus changing the fluid balance in the organism as well as other related issues - hepatorenal syndrome).
Otherwise the paper is good and I am happy to endorse it after receiving authors comments. Thank you.
Author Response
We would like to thank the reviewer for careful and thorough reading of this manuscript and for the thoughtful comments and constructive suggestions, which help to improve the quality of this manuscript. Our response follows:
Reviewer 3
Comment 1. In addition, advanced and metastatic HCC is usually accompanied by cachexia. Cachexia itself is a depletion tool for various compartments and impacting to decreased body composition. And the more advanced tumour the more profound is weight loss (at least in case of HCC). Could you please add a few sentences if there are any predisposing genetic differences for cachexia development in some patients.
Answer: Thank you very much for your suggestion. In the current manuscript, we aimed to discuss the impact of different body composition profiles on the outcome of HCC patients irrespective of the therapy. Due to the complexity of the pathophysiological mechanisms involved in the occurrence of sarcopenia/cachexia we would prefer this aspect to be mentioned very briefly in our manuscript. As you suggested we added some information concerning the impact of different genes activation in oncological patients. The added paragraph is written, as follows
On the other side the immune system via pro inflammatory cytokine (i.e. tumor necrosis factor alpha (TNF), IL-6, and transforming growth factor ß (TGF ß)) activate the expression of several genes, such as E3 ubiquitin ligase muscle RING finger containing protein 1 (MURF1) and muscle atrophy F box protein (MAFbx) , which are responsible for the catabolic state [22].
Comment 2 : Also, the advanced stages of liver cancer can be also associated with jaundice, ascites, transaminitis. Please add a few statements regarding their role in body composition balance shifting (e.g. bilirubin as a product of heme destruction is toxic if increased in circulation leads to possible renal damage leading to lower filtration thus changing the fluid balance in the organism as well as other related issues - hepatorenal syndrome).
Answer: Thank you very much for your comment. In the current manuscript, we aimed to discuss the impact of different body composition profiles on the outcome of HCC patients irrespective of the therapy and not to detail the pathophysiological mechanisms responsible for sarcopenia. We agree that this aspect is are critical; however, due to the subject's complexity, it requires a dedicated paper.
Concerning the impact of bilirubin on muscle, in theory, the serum bilirubin concentrations in sarcopenic patients are expected to be higher than those of non-sarcopenic patients. However, some authors found that bilirubin was a protective factor. In contrast, other authors found no association between TBIL and sarcopenia in patients with chronic liver diseases after adjustment for multiple potential confounding factors. Thus, due to these reasons, we did not include any information concerning BR in our manuscript.
References:
Kawamoto R, Ninomiya D, Kumagi T. Handgrip Strength Is Positively Associated with Mildly Elevated Serum Bilirubin Levels among Community-Dwelling Adults. Tohoku J Exp Med 2016; 240: 221–226.
Hyun Kim K, Kyung Kim B, Yong Park J, et al. Sarcopenia assessed using bioimpedance analysis is associated independently with significant liver fibrosis in patients with chronic liver diseases. Eur J Gastroenterol Hepatol 2020; 32: 58–65.
Wang C, Jin C, Yin X, Liu J, Liu J. Relationship between serum bilirubin concentration and sarcopenia in patients with type 2 diabetes: a cross-sectional study. J Int Med Res. 2021;49(3):3000605211004226. doi:10.1177/03000605211004226.
Round 2
Reviewer 1 Report
I have no other comments.